# LOW-RANK TENSOR ENCODING MODELS DECOMPOSE NATURAL SPEECH COMPREHENSION PROCESSES

## ABSTRACT

How does the brain process language over time? Research suggests that natural human language is processed hierarchically across brain regions over time. However, attempts to characterize this computation have thus far been limited to tightly controlled experimental settings that capture only a coarse picture of the brain dynamics underlying human natural language comprehension. The recent emergence of LLM encoding models promises a new avenue to discover and characterize rich semantic information in the brain, yet interpretable methods for linking information in LLMs to language processing over time are limited. In this work, we develop a low-rank tensor regression method to decompose LLM encoding models into interpretable components of semantics, time, and brain region activation, and apply the method to a Magnetoencephalography (MEG) dataset in which subjects listened to narrative stories. With only a few components, we show improved performance compared to a standard ridge regression encoding model, suggesting the low-rank models provide a good inductive bias for language encoding. In addition, our method discovers a diverse spectrum of interpretable response components that are sensitive to a rich set of low-level and semantic language features, showing that our method is able to separate distinct language processing features in neural signals. After controlling for low-level audio and sentence features, we demonstrate better capture of semantic features. Through use of low-rank tensor encoding models we are able to decompose neural responses to language features, showing improved encoding performance and interpretable processing components, suggesting our method as a useful tool for uncovering language processes in naturalistic settings.

## 1 INTRODUCTION

Natural language processing happens over time and regions in the brain, from early auditory processing to word parsing to semantics. Characterizing how language is processed has largely been carried out using non-invasive recording modalities namely, fMRI, EEG and MEG. Of these methods, MEG boasts a better temporal resolution than fMRI and better spatial resolution than EEG, making it an ideal candidate to study the fast processing of language. Classical literature has found stereotyped responses in brain activity to language in highly controlled experiments, such as the N400 response (Kutas & Federmeier, 2011). However, these experiments typically only focus on one aspect of language processing at a time and might not reflect how the system behaves in naturalistic settings.

As an alternative to the highly controlled setting, language experiments with naturalistic stimuli have become increasingly common in MEG (Wehbe et al., 2014b; Caucheteux & King, 2022; Gwilliams et al., 2024). To analyze these complex data, researchers have paired encoding models with word and sequence embeddings, often derived from language models. Recent research has identified a rich relationship between the processing of large language models (LLMs) and natural language processing in the human brain. Advances in LLMs have proven to be a source of rich language features that can successfully predict human brain responses (Wehbe et al., 2014b; Jain & Huth, 2018; Toneva & Wehbe, 2019; Schrimpf et al., 2021; Caucheteux & King, 2022). Conversely, the human brain has also served as inspiration for language and speech models, demonstrating an ability to drive their improvement in some cases (Toneva & Wehbe, 2019; Moussa et al., 2024; Vattikonda et al., 2025; Freteault et al., 2025). However, due to the lack of interpretability of the complex embeddings of language models, it is not simple to make sense of these correspondences.

Further, because the stimuli are not controlled by design, different low-level features can be correlated with semantic features represented in the embeddings, leading to spurious language model performance. Thus, it is difficult to determine which language features in an LLM lead to good predictions of brain activity. Recent research has suggested that language model features can be confounded with low-level features and simple model architecture changes such as position encoding (Hadidi et al., 2025). This highlights the need for new encoding model methods to be developed that can better disentangle the features of language model components contributing to brain activity, to better separate low level confounds from meaningful language processing.

In this paper, we present a low-rank tensor encoding model as a powerful method for predicting neural responses to naturalistic stimuli. Specifically, when applied to MEG recordings of participants listening to narrative audio, we simultaneously demonstrate improved encoding performance and interpretability over standard encoding model methods. The novelty of our work lies in applying low-rank tensor methods to MEG encoding models (which has not been done before) and demonstrating its ability to unlock new, interpretable and characterizations of neural activity as well as drive performance.

## 2 RELATED WORK

**Encoding Models**   Encoding models are defined as a map from a of feature set of stimuli to brain activity (often restricted to be linear) with the goal of predicting neural activity (Naselaris et al., 2011; Wang et al., 2025). These models have been used in a variety of contexts and modalities, such as vision (Kay et al., 2008; Nishimoto et al., 2011; Yamins & DiCarlo, 2016; Schrimpf et al., 2018; Lescroart & Gallant, 2019; Wang et al., 2023), audition (Kell et al., 2018), and language (Mitchell et al., 2008; Wehbe et al., 2014a; Huth et al., 2016). For modeling language responses, a popular and performant choice of features is to use a pretrained neural network language model (Wehbe et al., 2014b; Jain & Huth, 2018; Toneva & Wehbe, 2019; Schrimpf et al., 2021; Caucheteux & King, 2022), which host a battery of rich features useful for predicting a range of brain measurements including MEG (Wehbe et al., 2014b; Toneva & Wehbe, 2019; Caucheteux & King, 2022). These learned embeddings enable a new paradigm of neuroscience, allowing models to effectively describe neural activity during naturalistic behaviors and generalize outside of trained stimuli, motivating their use as digital twins (Jain et al., 2024; Wang et al., 2025). However, interpreting encoding models is fraught with difficulty. One salient challenge is how to separate low-level and high-level processing features, which limits their use in providing neuroscientifically useful results (Hadidi et al., 2025). Our work provides a new method for building encoding models that naturally separates low-level and high level language processing using low rank relationships over time and sensors.

**Encoding Model Interpretation Methods**   Understanding which stimuli features contribute to brain prediction performance in complex encoding models can be difficult. One method is variance partitioning, which can quantify the unique variance predicted by different feature groups (Borcard et al., 1992; Lescroart et al., 2015; de Heer et al., 2017; Lin et al., 2024), but it relies on knowing the groups of features in advance. One common method of characterizing an encoding model is to find the most activating stimuli for a particular brain region. This method has shown a degree of generalization, correctly predicting stimuli outside of the original stimulus set that when presented back to the brain strongly drive neural responses (Lorenz et al., 2016; Bashivan et al., 2019; Walker et al., 2019; Gu et al., 2022). An alternative method is to examine the weights on which features best predict neural activity. In fMRI, this approach has successfully identified semantic areas of the brain after projecting encoding model weights on their top principal components (Huth et al., 2016; Wang et al., 2023) or by using matrix decomposition on the brain data (Khosla et al., 2022). Comparatively less interpretation work has been done with encoding models for MEG (Caucheteux & King, 2022; Toneva et al., 2022; Gwilliams et al., 2024), which can deliver important knowledge about the dynamics of language processes

**Low-Rank Regression Models**   Tensor regression models are a natural extension to linear regression when independent and/or dependent variables have a multilinear structure. Similarly there exist tensor extensions of low-rank regression to different notions of low rank tensors, such as restricting the Tucker Tensor decomposition or Canonical Polydiadic (CP) Tensor decomposition (Rabusseau & Kadri, 2016; Lock, 2018). Neural activity has been found to lie on low dimensional subspaces

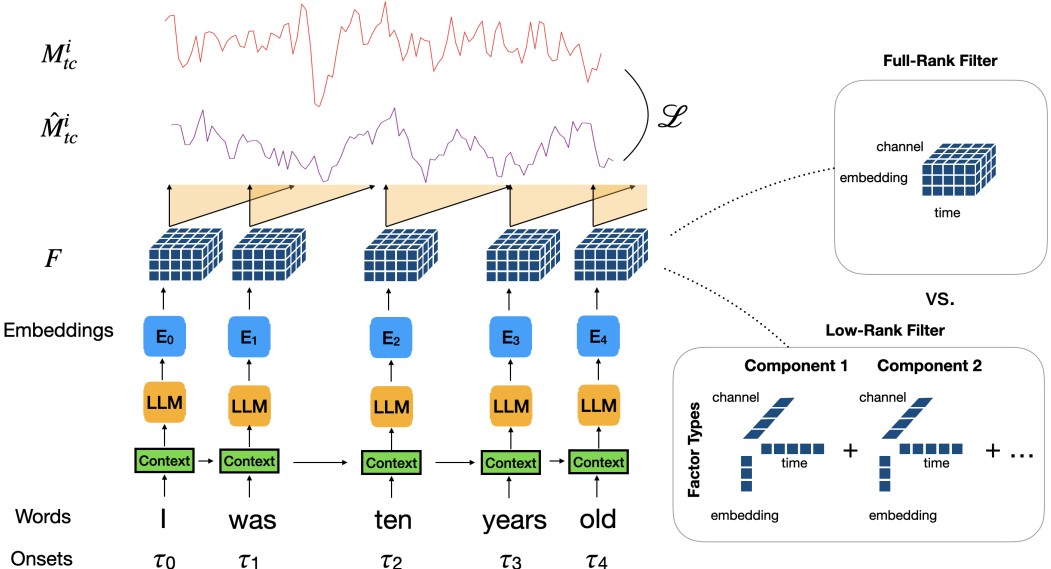

Figure 1: Low-rank tensor encoding model predicts brain activity along distinct rank-1 filter weight components over time, channels and language model embeddings. The previous 20 words provide context for the LLM embeddings, and the word onsets give the timing of the embedding, giving a high-dimensional time series. The low rank weight filters are then convolved with this time series and summed over components to generate the prediction of each channel over time.

in a variety of experimental settings and across brain areas (Churchland et al., 2012; Cunningham & Yu, 2014). This has motivated the use of low rank methods to find structure in neural activity, including tensor decompositions and tensor regressions (Williams et al., 2018; Pellegrino et al., 2024). Low-rank tensor regressions have also been used in neuroscience for regression problems (Ahmed et al., 2020; Zhou et al., 2013). While low-rank *matrix* encoding models have been used in language neuroscience (Vattikonda et al., 2025) to facilitate training (without interpretation), low-rank *tensor* methods have so far not been explored.

## 2.1 LOW-RANK TENSOR ENCODING MODEL

We define a time-delay encoding model on MEG as the following: Let the data of each story $i \in I$ be represented as the MEG signal $M^i \in \mathbb{R}^{T^i \times c}$ collected over $T^i$ time points and $c$ channels, with an additional tuple set $\{(w_j^i, \tau_j^i) \mid j \in [1, 2, ..., S^i]\}$ denoting the story words, $w_j^i$, spoken at corresponding time, $\tau_j^i \in [1, 2, ..., T^i]$ with a total number of words $S^i$ occurring in each story. For each word, the LLMs provide an embedding $E_j^i \in \mathbb{R}^N$. We construct our embedding time-series dataset as zero everywhere except at word times $\tau_j^i$:

$$X_t^i = \begin{cases} E_j^i & \text{if } t = \tau_j^i \\ \mathbf{0} & \text{otherwise.} \end{cases}$$

Unlike fMRI, whose time course is slower than the usual pace of spoken words, MEG is fast enough to resolve brain activity during the processing of each word. To transform from word embeddings to our MEG signal, we use a linear 3D FIR filter model with $D$ time delays and $C$ MEG sensors. This filter $F \in \mathbb{R}^{D \times N \times C}$ is convolved with the time series defined by the embeddings and their timings, and predicts activity at every sensor. This gives us a linear encoding model for predicting MEG responses $\hat{M}$ where

$$\hat{M}_{tc}^i = \sum_{d=0}^{D} \sum_{n=0}^{N} F_{d,n,c} X_{t-d,n}^i \tag{1}$$

To build a low rank tensor encoding model, we construct $F$ as a low rank Canonical Polydiadic (CP) tensor. We define a tensor filter of rank $R$ as the sum of $R$ rank-1 filter components, each

given by a product of three factors selected as rows of the matrices $U^D \in \mathbb{R}^{R \times D}, U^E \in \mathbb{R}^{R \times N}$, and $U^C \in \mathbb{R}^{R \times C}$ denoting time delay factors, embedding factors, and channel factors (see Figure 1):

$$F_{dnc}^R = \sum_{r=1}^{R} \alpha_r \, U_{rd}^D U_{rn}^E U_{rc}^C. \tag{2}$$

Each factor is normalized to length 1; all scaling is absorbed into one $\alpha_r$ per component $r$.

## 2.2 MODEL FITTING

We define our training loss on the MEG training data to be the mean squared error (MSE), with an optional ridge regularization parameter $\lambda_c$ for each channel to account for per-channel differences in signal-to-noise:

$$\mathcal{L} = \left[ \frac{1}{ITC} \sum_{i=1}^{I} \sum_{t=1}^{T_i} \sum_{c=1}^{C} (\hat{M}_{tc}^i - M_{tc}^i)^2 \right] + \frac{1}{CDN} \sum_{c=1}^{C} \lambda_c \sum_{d=1}^{D} \sum_{n=1}^{N} F_{dnc}^2.$$

where $\hat{M}$ is given by Equation 1.

To fit the full-rank tensor model, we used ridge regression with channel-specific penalties selected by 6-fold cross-validation over a log-spaced grid $[10^{-3}, 10^4]$. Folds were contiguous, equal-length segments within each story to respect temporal correlations and keep training/validation disjoint.

For the low-rank tensor model, we trained to convergence with stochastic gradient descent using Adam (lr $= 5 \times 10^{-3}$, batch size $= 300{,}000$) from random initialization. Factors were not constrained to unit norm, normalization constants were extracted post hoc. Because channels are coupled, a per-channel ridge search is combinatorial and infeasible. We therefore fixed the ridge penalty to 0.1 for all channels, near the full regression average. Models were implemented in PyTorch (Paszke, 2019) and run on 20 CPUs (512 GB RAM) with an L40 or A6000 GPU; each run took 3 h ($\approx$200 GPU-hours for results + $\approx$80 h exploratory).

In line with previous research, we used a 20-word context window (Toneva et al., 2022) and 40 time delays (0–800 ms at 50 Hz), where language-evoked MEG responses are strongest (Toneva et al., 2022).

Embeddings came from Llama-2-7B, layer 3 (Zhou et al., 2024). To reduce dimensionality, we applied PCA to the 4096-d embeddings and retained 665 components explaining 95% of training variance A1, leveraging the low-dimensional structure of LLM embeddings (Ethayarajh, 2019).

Performance on the held-out repeated test story was assessed with

$$CC_{\text{norm}}(c) = \frac{\text{Corr}\left( \bar{M}_{tc}^{\text{test}}, \hat{M}_{tc}^{\text{test}} \right)}{CC_{\text{max}}(c)}$$

where $\bar{M}^{\text{test}} = \frac{1}{|\text{test}|} \sum_{i \in \text{test}} M^{\text{test}}$ averages the five repeats and $CC_{\text{max}}(c)$ is the channel-specific noise ceiling (Schoppe et al., 2016; Antonello et al., 2023). We set a minimum noise-ceiling threshold of 0.2 to exclude low-signal channels.

## 2.3 MEG DATASET AND PREPROCESSING

To understand the processing of natural language over time, we chose to use an MEG dataset with natural language stimuli but also natural timing. In this experimental setup, subjects listened to naturalistic stories in the form of podcasts from the Moth Radio Hour. These stories have already been studied in fMRI (LeBel et al., 2023), and have been later scanned in MEG by an anonymous research group. In total, datasets for 3 subjects were collected using 27 unique stories over 5 sessions. Within each session, one story was repeated twice. Additionally, one story was repeated across all sessions. The data was collected in a MEGIN scanner using 306 channels with 102 cranial points at 1khz. Participants gave their informed consent and were remunerated for their time, and the study was approved by an anonymous institutional review board.

For MEG scan preprocessing, external sources were removed from the MEG signal with Signal Space Separation filtering, and a notch filter removed 60Hz signal from the power lines. The data

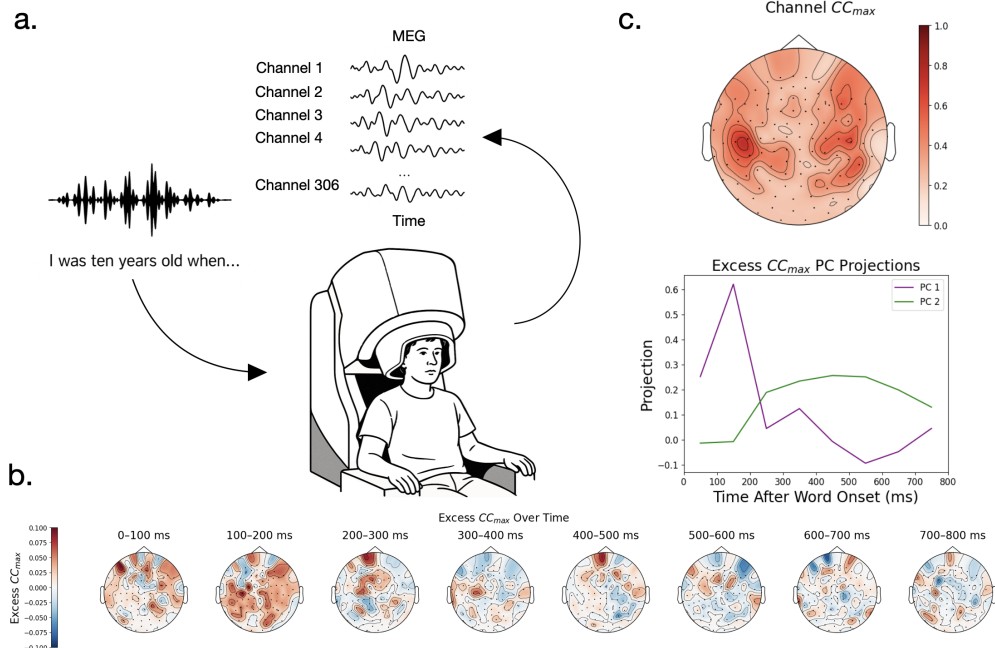

Figure 2: a) Subjects listened to Moth Radio Hour stories in an MEG scanner. The rest of the plots are shown for subject A. b) The predictability after word onset minus the average predictability (excess $CC_{\max}$) estimates over channels show distinct timecourses, responding strongly at 100-200ms and again more weakly at 400-500ms. c) Top panel: Neural activity shows higher predictability around the auditory cortex and language processing regions. Bottom panel: Projecting the excess $CC_{\max}$ onto its first two Principal Components (PCs) over channels reveals that after the auditory onset response, neural activity should be predictable on the timescale of semantic processing.

was then band pass filtered between 1hz and 150hz. Finally, ICA was run to remove any sources highly correlated to eyeblinks or the raw audio signal. During initial analysis, we noticed that the audio was stretched by a small (imperceptible) amount while being played in the scanner. This stretch was approximately consistent within each story across subjects; we corrected for this effect by resampling. All analysis was performed on a version of the MEG signal that we downsampled to 50hz for computational efficiency. In total our dataset comprised of approximately 58,000 words (excluding the words from repeated stories) and approximately 1.3 million MEG samples per subject.

The labels of words and their respective timings were used from LeBel et al. (2023). However, the words consisted of capitalized and un-punctuated words which are likely to be out of distribution for a language model. To build a transcript that falls into natural language, we used a strong language model, GPT-4, and prompted it to keep the transcript words the same but convert it into our desired format. After passing it through the language model, each transcript was human checked and repaired to ensure that each word remained the same.

For all training runs, we used the same dataset composed of all stories except for a held out test set of one new story with 5 repeats. This ensured no contamination between training and test. Before training, the MEG data was normalized per channel per story.

## 3    RESULTS

### 3.1    NEURAL PREDICTABILITY POST WORD ONSET

Before fitting encoding models to our dataset, we wanted to investigate the predictability of the neural signal across channels. To do this, we used neural responses to 5 repetitions of one story to compute the noise ceiling, $CC_{\max}$. Subjects displayed predictability in a variety of brain regions (Figure 2c, A2), with more predictability in auditory and language processing regions.

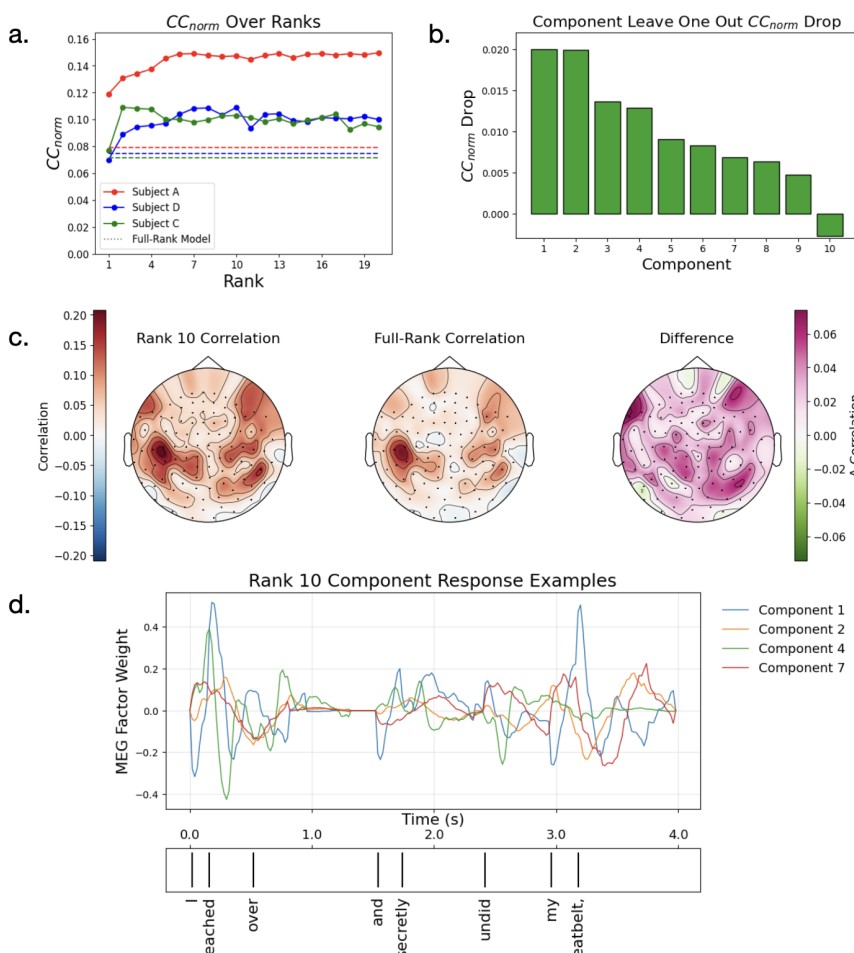

Figure 3: a) Low-rank tensor models improve encoding performance, measured by the channel average $CC_{\mathrm{norm}}$. Dotted line represents the performance of a full rank regression model for that subject. Low rank performance saturates early with a small number of components. b) Components contribute differently to encoding performance, measured by the drop in performance when omitting that component. c) Low-rank models improve encoding performance against full models broadly over channels (subject A). d) Timecourses of component weights: Each component is driven by a pattern of language features over time, and predicts a spatial pattern across channels according to $U_{rc}^{C}$ (Equation 1). These component weights ($W_{rt} = \alpha_r \sum_{n,d} U_{rd}^{D} U_{rn}^{E} X_{t-d,n}^{i}$) indicate the strength of each of these spatial patterns in the predicted MEG signal. These predicted brain responses to language captured by these components show a diversity of processing (subject A).

To see how the predictability of the signal changes as a function of time after word onset, we made subsets of our neural data over time and computed their noise ceiling. Each subset was constructed using a time window post-word onset of 100ms. These time windows were then concatenated together to form a subsampled time series which we used to compute the $CC_{\mathrm{max}}$. To view the change in the difference of $CC_{\mathrm{max}}$ over time, we subtract out the $CC_{\mathrm{max}}$ over all time points to get the excess $CC_{\mathrm{max}}$ given by that time window.

We find that the excess $CC_{\mathrm{max}}$ fluctuates over time and channels (Figure 2b for subject A, other in A3), with a distinct peak in 100-200ms for all subjects, likely corresponding to initial processing of the audio signal, and a shallower, broader peak again around 400ms in subjects A and D, likely corresponding to more semantic processing. Doing PCA on the excess $CC_{\mathrm{max}}$ of subject A makes these timescales more apparent. These analyses suggest that predictability of neural activity varies on multiple timescales post-word onset (Figure 2a).

## 3.2 Low-Rank Encoding Models Improve Performance

A low-rank encoding model carries an inductive bias that the data being modeled is low dimensional. This might be relevant to MEG data either because the underlying brain activity is low dimensional or because, due to physical limitations and distortions, the MEG sensors sample from a broad region of underlying cortex, thus leading to redundancies. To investigate whether the low-rank inductive bias is useful in predicting MEG activity during natural language processing, we fit a series of low-rank models with increasing rank (Figure 3a). We find that indeed low-rank models predict neural activity better, in some cases drastically such as Subject A, where the encoding performance $CC_{max}$ effectively doubles. To test the significance of this result, we perform a bootstrap hypothesis test against the performance of the full regression model as well as a permutation test against chance performance. We find all performance to be significant at 0.001 except for Subjects C and D's rank 1 models, which fail to significantly outperform regression (A12, A13). To see where low-rank models drive encoding performance, we took rank-10 models and compared their performance over channels to the full-rank model (Figure 3c, A4). We find that the difference is generally widespread, improving the predictions over many areas. These results suggest that low-rank models are more predictive than standard encoding models on our MEG dataset.

## 3.3 Low-Rank Components Capture a Diverse Spectrum of Neural Activity

We now ask what language processes the low-rank model is capturing. We find that low-rank encoding models learn diverse, complex activations over time in response to language features (Figure 3d, A5). However, their contributions may not be equally predictive of the MEG signal. To estimate the predictive power of a single component, we consider the drop in mean $CC_{\mathrm{norm}}$ when that component is excluded from prediction. We find that not all components contribute equally, with some having a much larger effect than others and, in rare cases, dropping components leading to improvements (Figure 3b, A6). For subsequent analysis, we sort components by their leave-one-out influence, starting with the most influential component.

Next, we study the factors individually. Since our factors are defined as having a vector norm of 1, with the norm value being absorbed into a single multiplicative coefficient $\alpha_r$, we have no scale ambiguity in our interpretations (Equation 2). We find that the coefficients are roughly the same across components even though they have different influence (A11, Figure 3b). To analyze the components, we examine a model with 10 components for all subjects. Within this low-rank model are a diverse set of processing time-courses and spatial modes (Figure 4, A7). Largely we find that the spatial modes sit over auditory and language areas (Figure 4, A8). The timecourses vary but many show peaks around 150–200ms and 300–400ms.

To understand the language features driving each component, we searched over the space of contexts from the Moth Radio Hour transcript corpus. MEG predictability saturates after just a few words of context, consistent with previous literature (Toneva et al., 2022) A9. Because of this, we examine the most activating contexts with 5 words. For each component we select the five most activating contexts from all transcribed stories as those whose LLM embedding have the highest dot products with that component's embedding factor $U_r^E$. Similarly, for the most negative activating contexts we select the most negative dot products. Examining these selected contexts, we find that most factors are activated by low level language features, such as end of sentence and start of sentence (Figure 4, factor 1). However, some respond to more semantic features (Figure 4 factor 7). To quantify the low rank features, we count the ratio of factors whose most activating (positive or negative) components respond to only starts of sentences or punctuation. We find that 8/10, 9/10 and 9/10 factors of subjects A, C and D satisfy this criteria.

In summary, the low-rank components capture a wide set of distinct language processes over time, channels, and language embeddings. These components are dominated by the encoding of low-level features but high level aspects also contribute to MEG prediction. In the next section, we control for the effect low-level features and show that the model can still predict high-level components in MEG.

Anecdotally, our model was quite useful at detecting an experimental artifact. In development, we noticed that some of our factors were nonsensical and showed high-frequency temporal fluctuations. This allowed us to diagnose some leakage of audio-signal into the MEG signal, which we later fixed in preprocessing. This example showcases the utility and power of our low-rank decomposition.

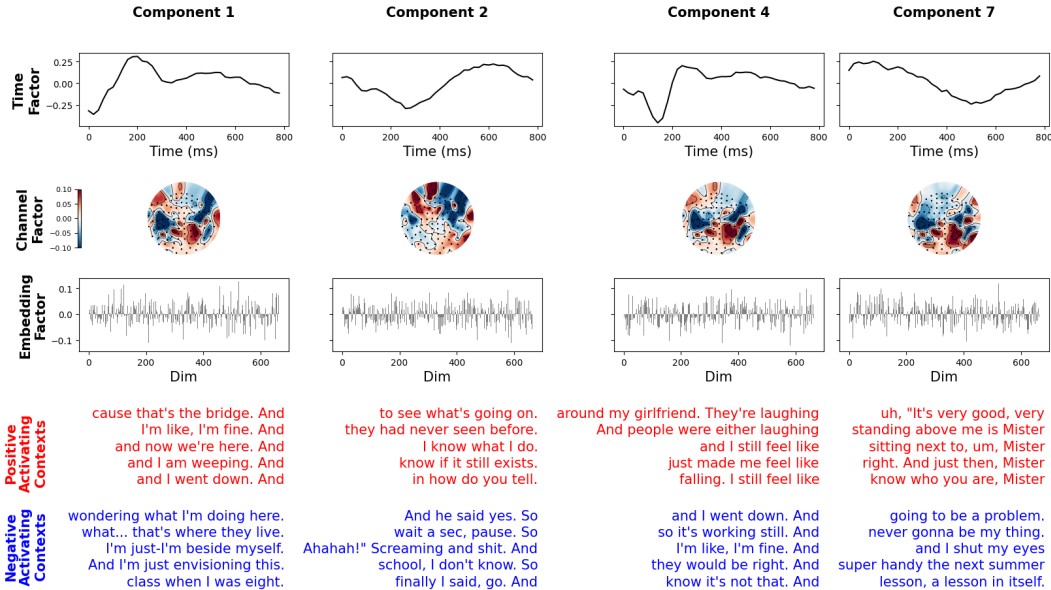

Figure 4: The low rank model decomposes neural activity into distinct 1-dimensional factors that depend on time, channels, and language embedding. *Top three rows*: Several example components' factors are shown from a rank-10 model of subject A, demonstrating the spectrum of different activations over time and channels. Several time factors and channel factors show distinct patterns, suggesting the decomposition is separating distinct aspects of language processing. *Bottom two rows*: Top 5 most positively (red) and negatively (blue) activating five-word contexts, for each component's language embedding factor. The factors are dominated by low level effects such as sentence start and end (components 1,2,4) while some capture more semantic information (component 7).

## 3.4 LOW-LEVEL CONTROLS IMPROVE SEMANTIC FEATURES

We previously found that the low-rank components are dominated by low-level effects. We want to see if subtracting out these low-level signals leads to any changes in the semantic content captured by the low rank factors. For our control set, we constructed a features space of a log-mel spectrogram, word onset and sentence start/end. To construct the log-mel spectrogram, we downsample each story audio signal to 16kHz with a hop length of 25ms and 80 Mel features over 15 time delays. We encode word onsets, sentence starts, and sentence ends each as binary sequences taking a value of 1 at the times the respective features appear, and 0 at all other times. We then fit a ridge-less full-rank model over the train and test set of the features and regressed out the control signals.

After controlling for low-level language properties, residual activity components change in three ways. First, their most-activating contexts become more semantic(Figure 5c, A10): only 3/10 remain low-level in subject A and 8/10 in C, while D remains mostly low-level (9/10). Second, a normalized power analysis shows spatial peaks shift from auditory regions to other brain areas (Figure 5b, A8,A15). Third, the same analysis shows later temporal peaks in A and D and double peaks in all, consistent with longer processing times for higher-level language comprehension.(Figure 5a, A5).

We assessed shared neural structure captured by the control-subtracted low-rank tensor models using PCA on each factor type to estimate subspace dimensionality. Within subjects, most variance lies in the first few dimensions. We define the cross subject factor space to be the concatenation of all factor matrices of a single type over the rank dimension. Across subjects, we built explained-variance curves with a ceiling and a baseline. The ceiling tiles one subject's factor matrix along the rank dimension to match the number of subjects. The baseline orthogonalizes each subject's factor matrix to the others before concatenation. PCA is then run on each cross-subject factor matrix. We quantify similarity as the area between the actual and baseline curves divided by the area between the ceiling and baseline. Cross-subject similarity was high for time and space factors (85% and 81%) and lower, though substantial, for embedding factors (32%). This suggests that the individual subject models are learning similar structures(see A16).

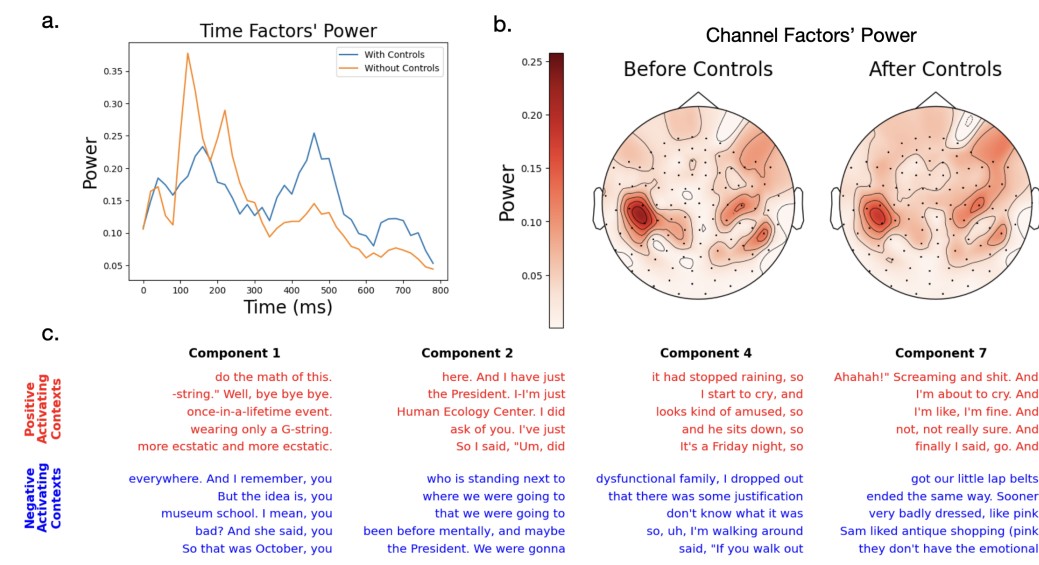

Figure 5: Low-rank model of residual responses after controlling for low-level language features, compared to a model applied to neural activity without controls. All panels shown for Subject A. a)Power in time factors shifts later in residual responses versus non-controlled data. Un-normalized power here is defined as $Q_t^D = \sum_r \alpha_r^2 (U_{rt}^D)^2$ and the power is $P^D = Q^D / ||Q^D||$. b) Power in channel factors becomes more distributed for control subtracted data versus non-controlled data. Un-normalized power here is given by $Q_c^C = \sum_r \alpha_r^2 (U_{rc}^C)^2$ and power is $P^C = Q^C / ||Q^C||$. c) Words that activate the factors' language embedding most (red) and least (blue), as in Figure 4. These contexts capture more semantic and less low-level features like ends of sentences.

After removing low level controls, we find more semantic features are captured in the low rank components. This indicates that though the presence of low-level confounds is a concern for understanding language model brain predictability, low-level features do not capture all model predicted variance. Our results suggest that future encoding model works should focus on capturing explainable variance after subtracting out low level controls to capture meaningful language processing.

## 4 CONCLUSION

In this work we predicted brain activity from language features using low-rank tensor encoding models. We showed that this method improves encoding performance over full rank linear regression models. Beyond this, we demonstrate that our model is able to naturally and effectively capture interpretable low- and high-level language processing features over time and space. Finally, by controlling for low-level features, we show we can isolate more semantic components.

While we applied this low-rank tensor encoding model to an MEG dataset, we see it as a general method useful for building interpretable and performant encoding models in neuroscience. We are excited about the application of this method to other encoding model modalities like video and audio as well as other high temporal resolution brain recording methods such as ECoG, EEG, and microelectrode recordings.

While our model is able to capture diverse language processing features in multiple subjects, our analysis was limited to a comparative study over each subject individually. To better capture the low-rank components shared across subjects, a single low-rank model could be used on all subjects, stacking MEG features over subjects listening to the same auditory stimuli. Additionally, in this paper, we subtract a limited number of controls from the MEG signal when investigating the model's learned semantic feature encoding. To improve the interpretations of the model factors, more rich control sets could be used. We leave both these as lines of investigation for future work.

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

## 5 ETHICS

Our human subject dataset satisfied all human data requirements such as consent, fair wage, IRB approval etc (section 2.3) and were anonymized. All subject data that will released is similarly be anonymized. Our work has minimal forseeable negative societal impact and potential harmful consequences as well.

## 6 REPRODUCIBILITY

We provide preprocessing steps, hyperparameters etc. used for fitting. Additionally, we provide code in the supplemental material and an anonymized data link, to be released publicly following potential acceptance. The human MEG experimental setup is discussed in section 2.3 as well as shown Figure 2. Finally, we reference the audio dataset used, allowing anyone to collect similar data to the MEG dataset used here. Our model is clearly explained in section 2 as well as figure 1.

## A APPENDIX

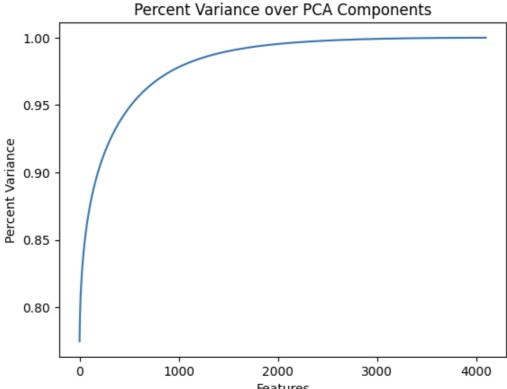

Figure A1: Embedding percent variance explained over PCA components of Llama2-7b on the Moth Radio Corpus. 95% of the variance is captured at a small percent of the total dimension.

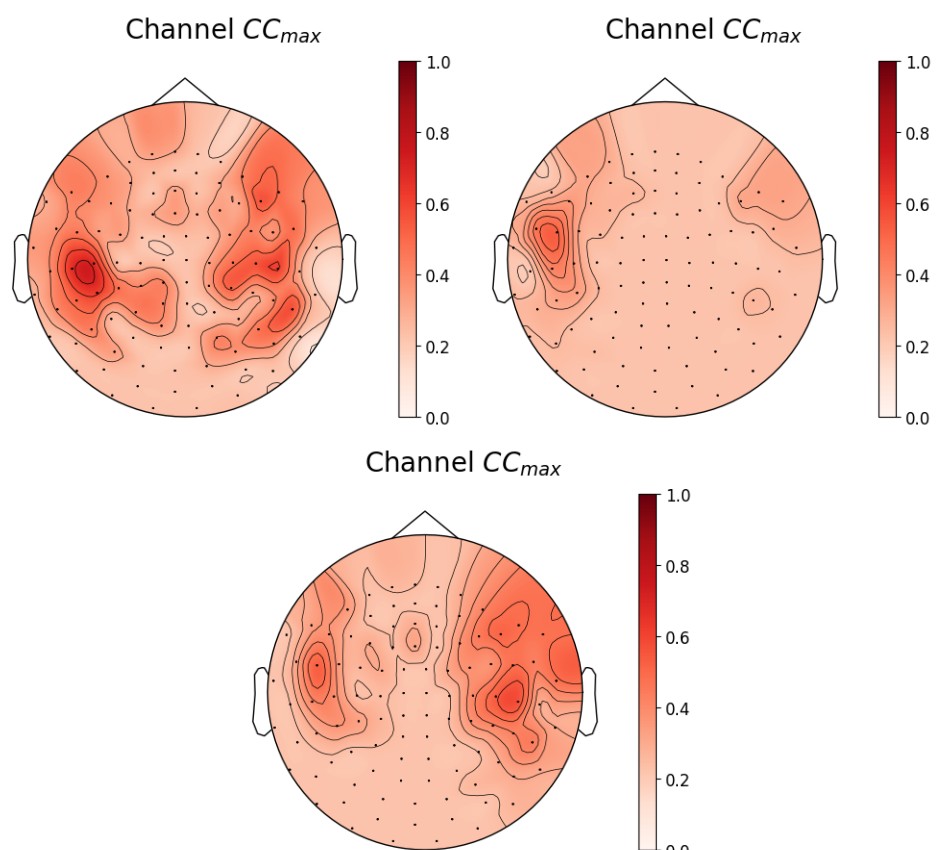

Figure A2: $CC_{max}$ over channels. Subjects A and D show much better predictability over channels. *Top Left*: Subject A. *Top Right*: Subject C. *Bottom*: Subject D.

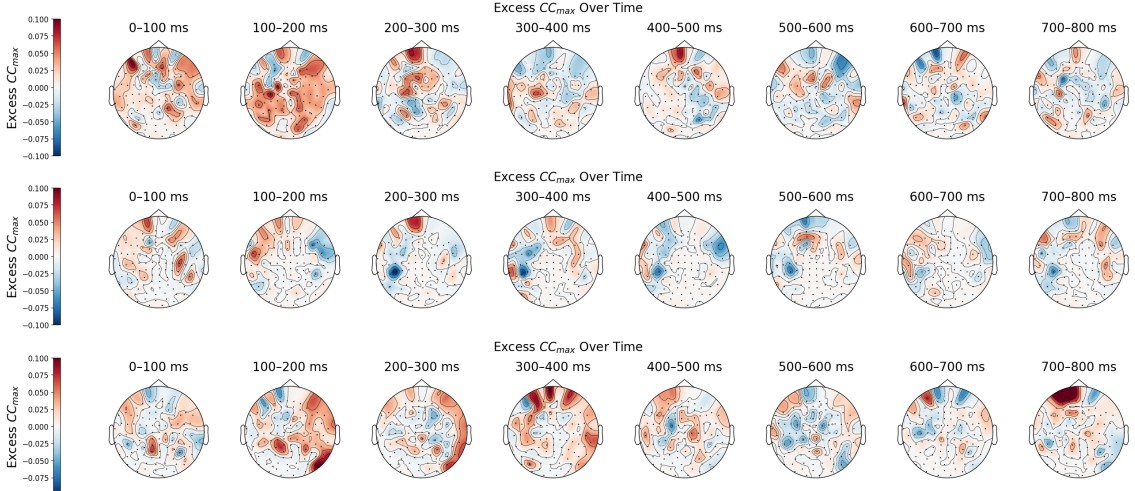

Figure A3: Channel excess $CC_{max}$ shows distinct patterns over time. Subjects A and D show early peaks in predictability around 100-200ms post word onset and an additional shallow peak at 400-500ms in subject A and 300-400ms in subject D. *Upper panel*: Subject A. *Middle panel*: Subject C. *Lower panel*: Subject D.

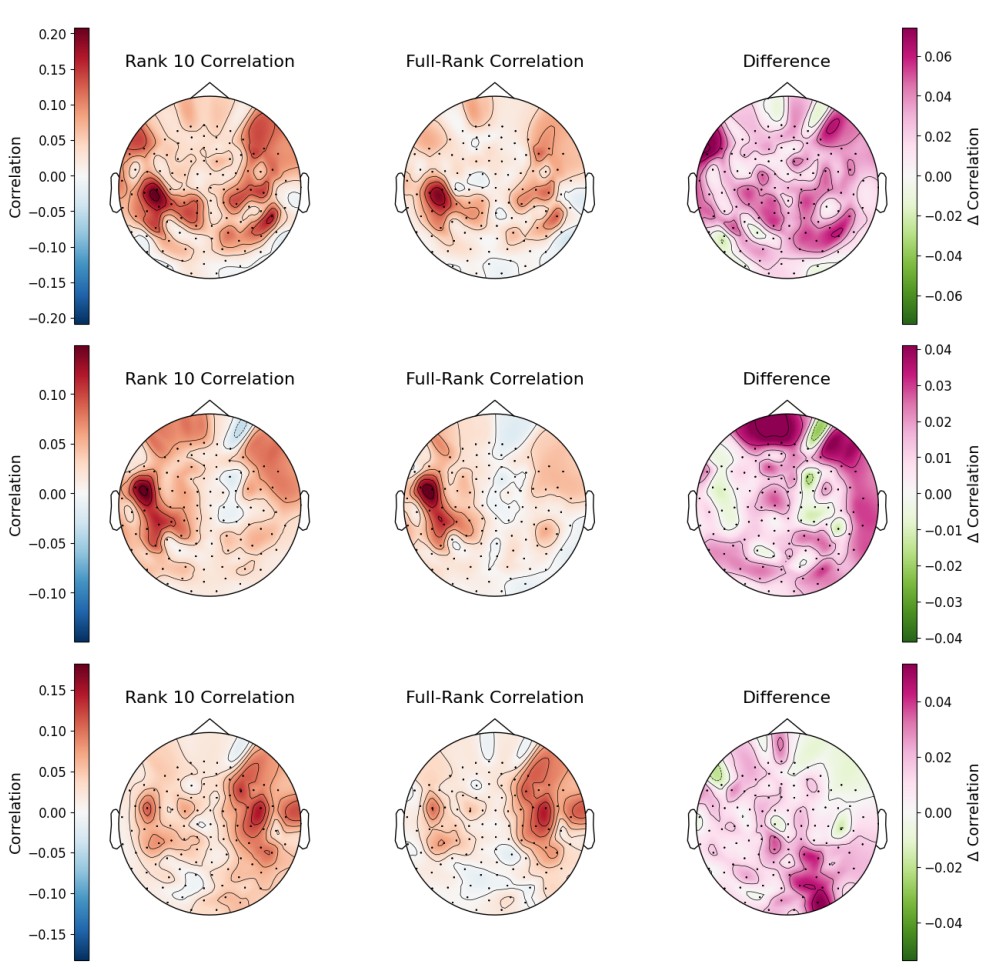

Figure A4: Rank 10 models show widespread performance improvements over full ridge regression.*Top panel*: Subject A. *Middle panel*: Subject C. *Bottom panel*: Subject D.

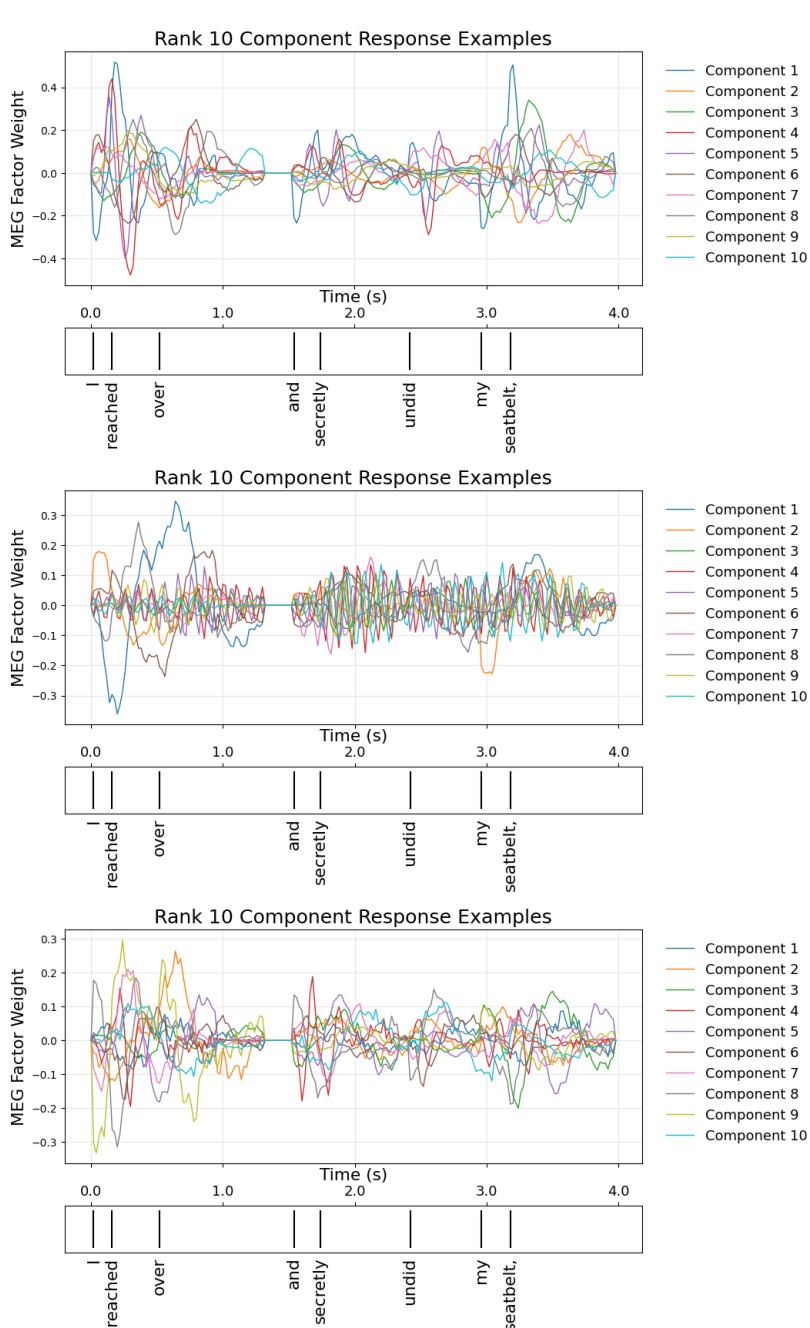

Figure A5: Timecourses of components pre-MEG factor multiplication. Factors show a diversity of timecourses in response to natural language. *Top panel*: Subject A. *Middle panel*: Subject C. Note the very fast components, likely corresponding to the small amount of signal in subject C compared to the number of ranks. *Bottom panel*: Subject D.

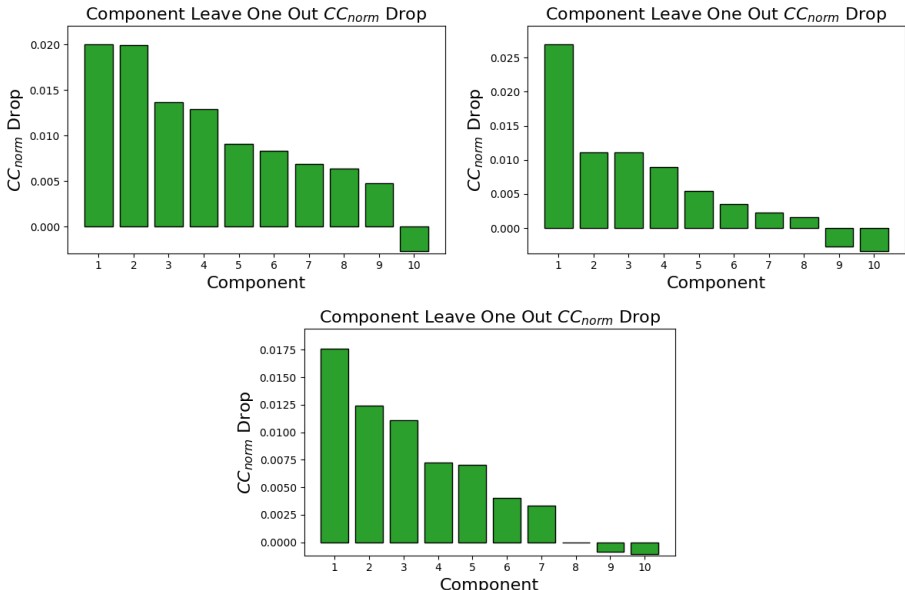

Figure A6: Leave one component out drop in $CC_{norm}$ over rank. Components show different influence on test performance. *Top Left*: Subject A. *Top Right*: Subject C. *Bottom*: Subject D.

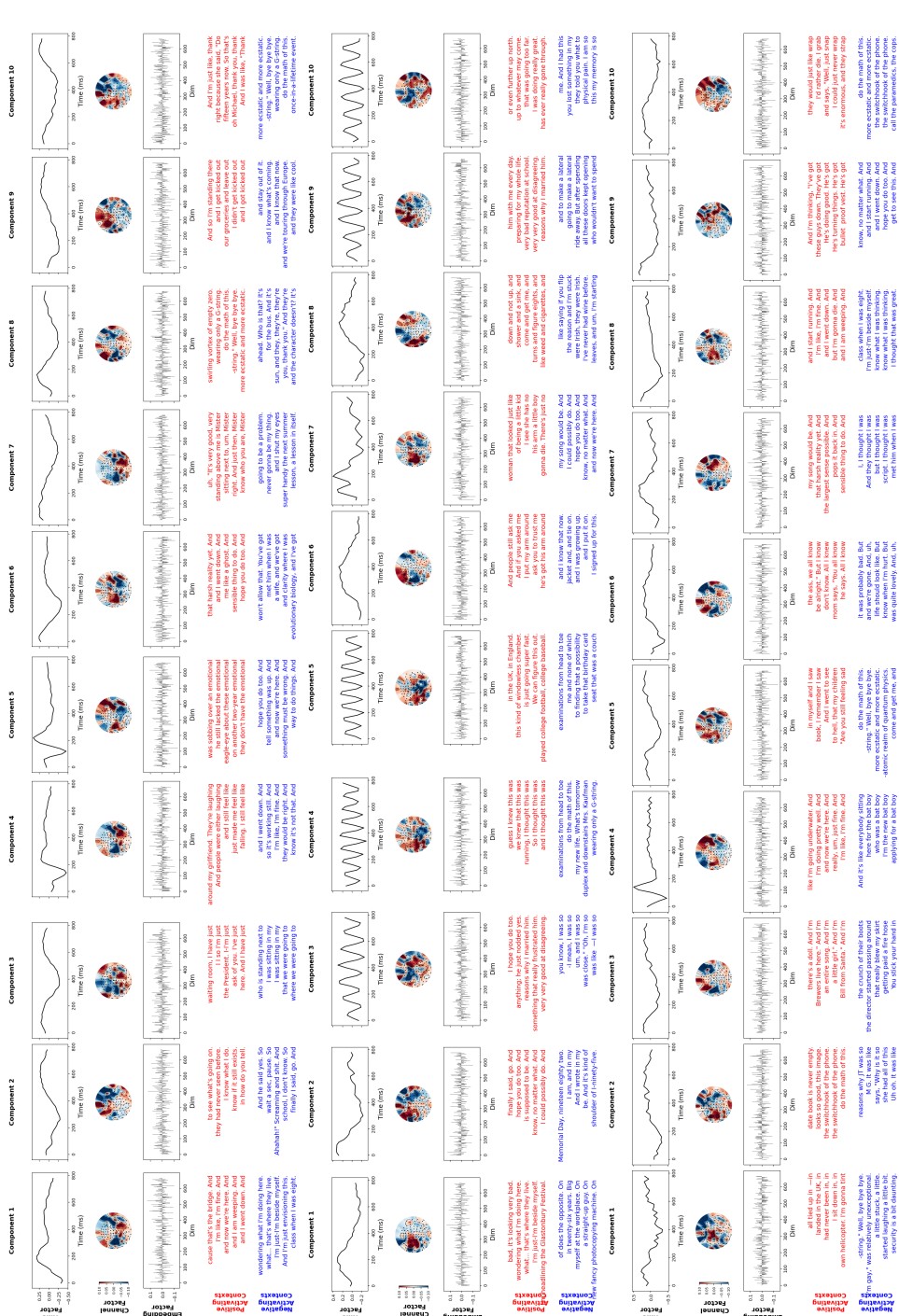

Figure A7: Subjects show a diversity of timecourses, spatial components, and activating texts. *Left panel*: Subject A. *Middle panel*: Subject C. Note the fast timecourse components, likely stemming from low signal in the subject. *Right panel*: Subject D.

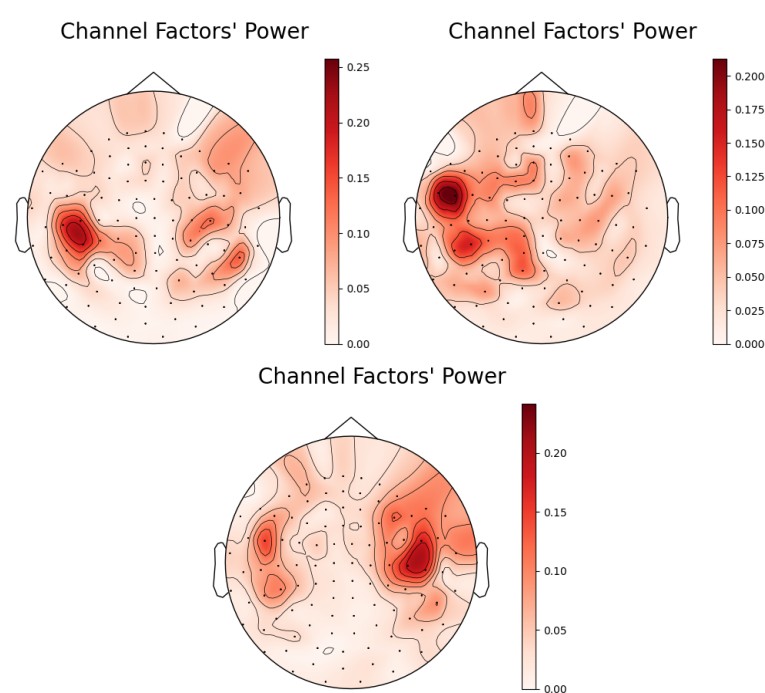

Figure A8: Using the same power analysis as main figure 5 shows that subject channel factors concentrate over auditory areas (left and right hemisphere, roughly in temporal and prefrontal regions).*Top Left*: Subject A. *Top Right*: Subject C. *Bottom*: Subject D.

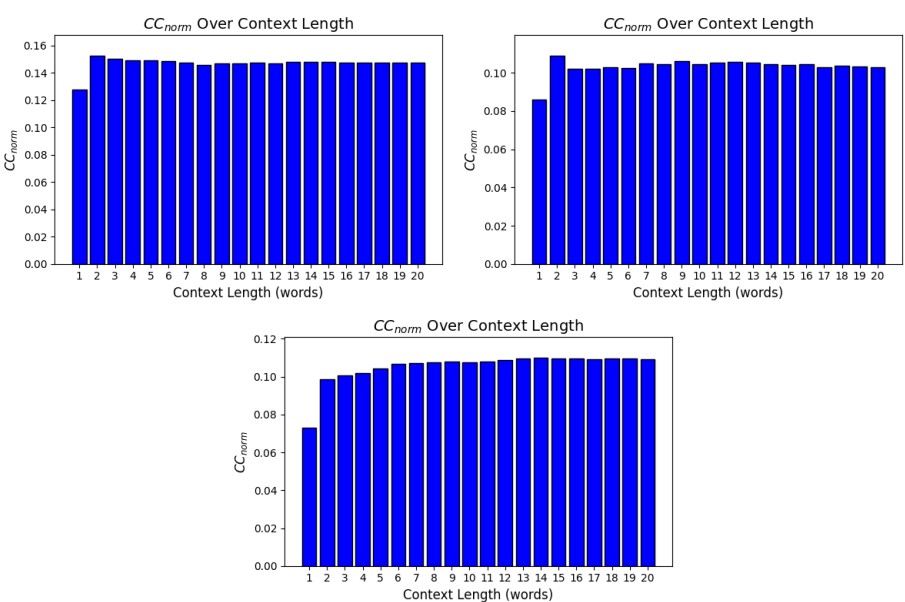

Figure A9: $CC_{norm}$ saturates after a small number of previous words being included in the context when evaluated on a rank-10 model trained on a 20 word context window. *Top Left*: Subject A. *Top Right*: Subject C. *Bottom*: Subject D.

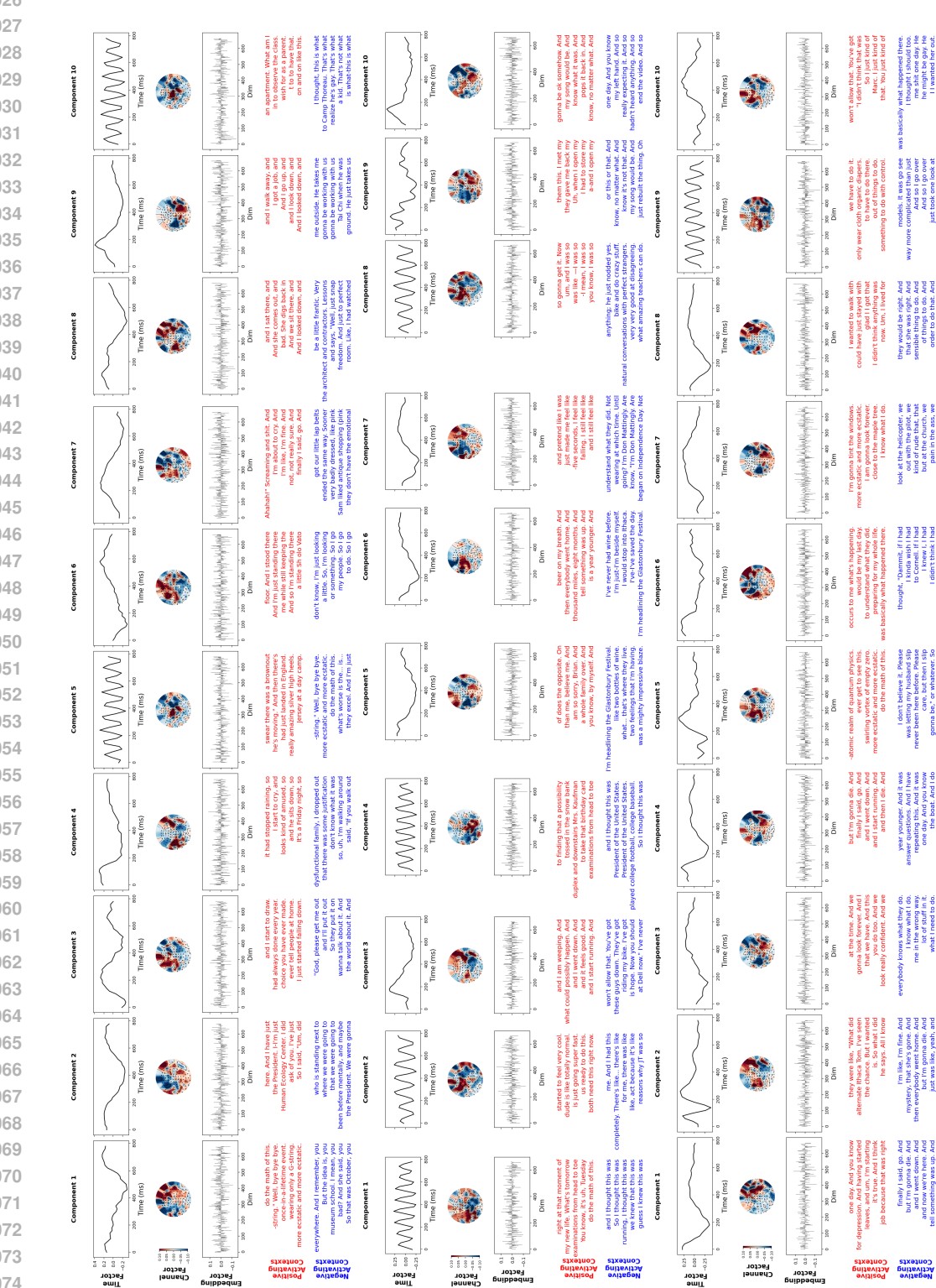

Figure A10: Control subtracted factors for a rank-10 model. We see changes in timecourses, channels and activating texts from the un-controlled rank-10 model. Note the fast timecourses, likely arising from less signal in control subtracted residuals. *Left panel*: Subject A. *Middle panel*: Subject C. *Right panel*: Subject D.

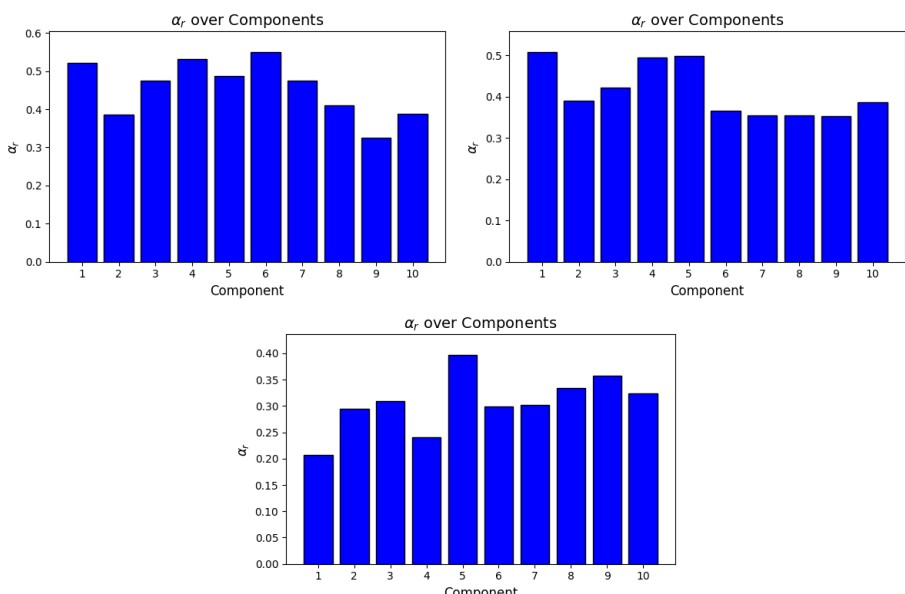

Figure A11: Component scalings show roughly similar ranges over components despite having different influences on test performance (see Supplemental Figure 6).*Top Left*: Subject A. *Top Right*: Subject C. *Bottom*: Subject D.

| Subject | Rank 1 | Rank 2 | Rank 3 | Rank 4 | Rank 5 | Rank 6 | Rank 7 | Rank 8 | Rank 9 | Rank 10 |
|---------|--------|--------|--------|--------|--------|--------|--------|--------|--------|---------|
| A | 0.001 | 0.001 | 0.001 | 0.001 | 0.001 | 0.001 | 0.001 | 0.001 | 0.001 | 0.001 |
| C | 0.208 | 0.001 | 0.001 | 0.001 | 0.001 | 0.001 | 0.001 | 0.001 | 0.001 | 0.001 |
| D | 0.783 | 0.001 | 0.001 | 0.001 | 0.001 | 0.001 | 0.001 | 0.001 | 0.001 | 0.001 |

| Subject | Rank 11 | Rank 12 | Rank 13 | Rank 14 | Rank 15 | Rank 16 | Rank 17 | Rank 18 | Rank 19 | Rank 20 |
|---------|---------|---------|---------|---------|---------|---------|---------|---------|---------|---------|
| A | 0.001 | 0.001 | 0.001 | 0.001 | 0.001 | 0.001 | 0.001 | 0.001 | 0.001 | 0.001 |
| C | 0.001 | 0.001 | 0.001 | 0.001 | 0.001 | 0.001 | 0.001 | 0.001 | 0.001 | 0.001 |
| D | 0.001 | 0.001 | 0.001 | 0.001 | 0.001 | 0.001 | 0.001 | 0.001 | 0.001 | 0.001 |

Figure A12: $CC_{norm}$ differences between low-rank models vs full linear rank regression are significant at 0.001 for all models except for Subject C and D rank-1 models. Shown above are the estimated p-values from a bootstrapped test performance measure. To estimate the p-value, test data was split into sequential blocks of 10 seconds, sampled from with replacement and used to construct a new test dataset. The performance difference between the low-rank model and full rank model were calculated for each new test set. The p-value is then the ratio of performances less than 0 to total bootstraps, in this case 1000

| Subject | Rank 1 | Rank 2 | Rank 3 | Rank 4 | Rank 5 | Rank 6 | Rank 7 | Rank 8 | Rank 9 | Rank 10 |
|---------|--------|--------|--------|--------|--------|--------|--------|--------|--------|---------|
| A | 0.001 | 0.001 | 0.001 | 0.001 | 0.001 | 0.001 | 0.001 | 0.001 | 0.001 | 0.001 |
| C | 0.001 | 0.001 | 0.001 | 0.001 | 0.001 | 0.001 | 0.001 | 0.001 | 0.001 | 0.001 |
| D | 0.001 | 0.001 | 0.001 | 0.001 | 0.001 | 0.001 | 0.001 | 0.001 | 0.001 | 0.001 |

| Subject | Rank 11 | Rank 12 | Rank 13 | Rank 14 | Rank 15 | Rank 16 | Rank 17 | Rank 18 | Rank 19 | Rank 20 |
|---------|---------|---------|---------|---------|---------|---------|---------|---------|---------|---------|
| A | 0.001 | 0.001 | 0.001 | 0.001 | 0.001 | 0.001 | 0.001 | 0.001 | 0.001 | 0.001 |
| C | 0.001 | 0.001 | 0.001 | 0.001 | 0.001 | 0.001 | 0.001 | 0.001 | 0.001 | 0.001 |
| D | 0.001 | 0.001 | 0.001 | 0.001 | 0.001 | 0.001 | 0.001 | 0.001 | 0.001 | 0.001 |

Figure A13: $CC_{norm}$ performance is above chance for all models at 0.001. Shown above are the estimated p-values from a permuted test performance measure. To estimate p-values, we first split our test MEG data into sequential blocks of 10 seconds. Then for each permutation sample, we shuffled the blocks to create a new test set and computed the performance of the model on it. Using all permutations (in this case 1000), we construct the null distribution that our model is equal to a chance model. Then, we obtain the p-value as the probability that the null distribution is greater than the test performance of the model.

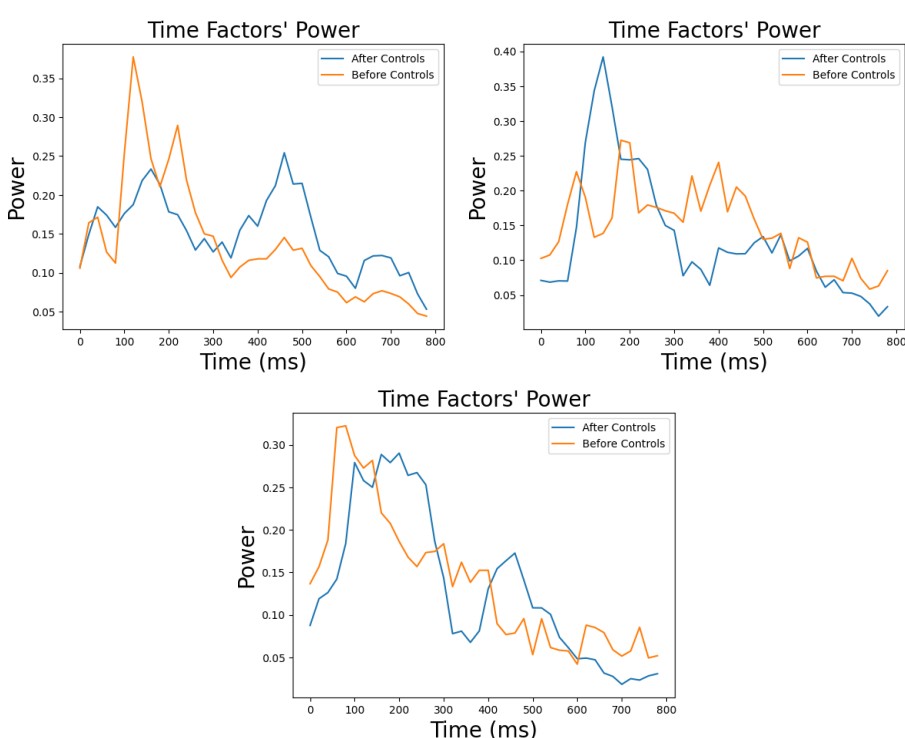

Figure A14: The power in the time factors shifts to be later in subjects A and D after subtracting controls. In subjects A and D, the power is approximately monotonic before subtracting controls, peaking at around 150ms. After control subtraction, double peaks form, one early peak at around 150-200ms followed by a later peak around 400-500ms. Subject C exhibits this double peak behavior, however after controls, it becomes more peaked around 150ms.*Top Left*: Subject A. *Top Right*: Subject C. *Bottom*: Subject D.

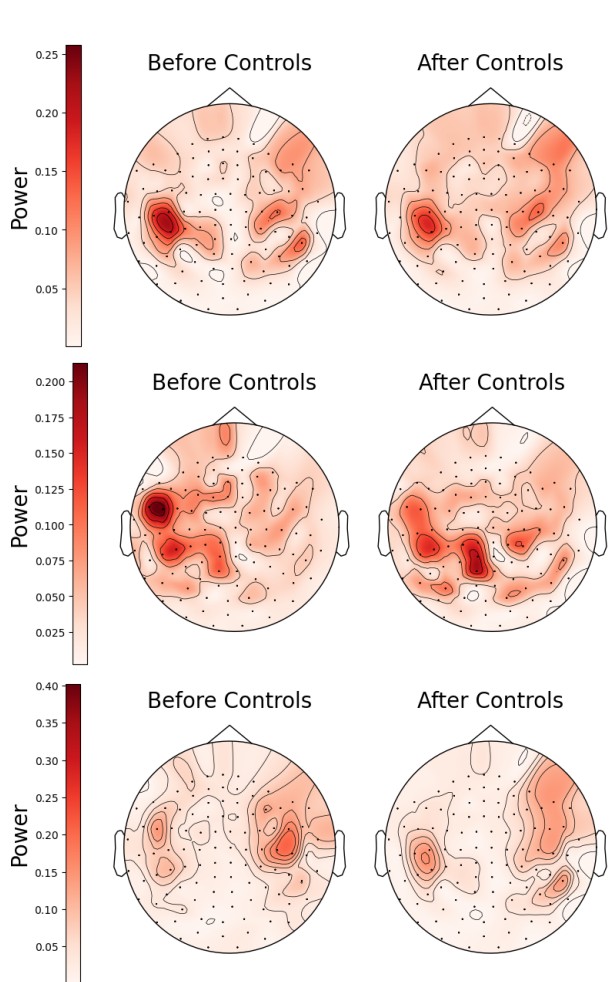

Figure A15: Power in channel factors before and after subtracting controls. Generally, across subjects, channel factors shift to become less concentrated over auditory areas, affecting channels more globally following control subtraction. *Top*: Subject A. *Middle*: Subject C. *Bottom*: Subject D.

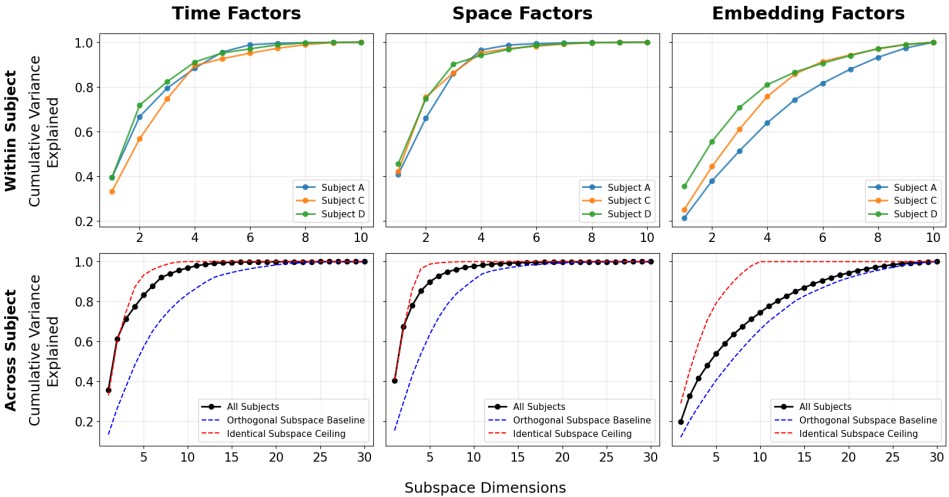

Figure A16: Factors lie in low dimensional spaces within and across subjects. *Top Panel*: Factors of each type within each subject lie in a low dimensional space with the majority of the cumulative variance explained occuring in the first few factors. *Bottom Panel*: Across subjects, the combined factors of each type also lie in a low dimensional space. To construct each cross-subject factor space, the factors from each subject were simply concatenated to eachother over the rank dimension before running PCA. The ceiling is calculated using a factor subspace created by replacing the factors of other subjects with a single subject's factors. The baseline is calculated by rotating the subspace basis vectors of each subject to be orthogonal across subjects while maintaining each subject's singular value distribution.

