# OpenReview forum: "Low-Rank Tensor Encoding Models Decompose Natural Speech Comprehension Processes"
_ICLR.cc/2026/Conference — Submitted to ICLR 2026_

### Official Review · Reviewer_n1HC · 2025-10-25

**Soundness:** 2
**Presentation:** 3
**Contribution:** 2
**Rating:** 4
**Confidence:** 5

**Summary:**

This work is part of a broader research effort to explore the relationship between internal representations of large language models and human brain activity recorded during speech comprehension. Specifically, the authors focus on building a new encoding model that uses low-rank tensor decomposition to interpret which stimulus features that truly predict human brain recordings (MEG) while participants listen to naturalistic The Moth Stories. The evaluation focuses on comparing the encoding performance of the new encoding model against the prior full-rank ridge regression model and demonstrates how the new encoding model effectively disentangles interpretable features (low-level vs. semantic) in language processing over time. The authors regress low-level control-driven MEG activity from original MEG activity, using the residual MEG low-rank tensor approach to examine whether more semantic related activity is observed over time. Overall, low-rank tensor neural encoding models provide an interpretable alternative approach for mapping language model embeddings to brain activity over time.

**Contributions:**

* *Introduction to low-rank tensor decomposition in an encoding model:* The study presents a low-rank tensor decomposition in neural encoding model for predicting MEG brain activity, which is conceptually novel.
* *Comprehensive evaluation:* The study compares the encoding performance of a low-rank tensor decomposition model and a full-rank ridge regression model. For each encoding model, the authors measure the correlation between actual and predicted brain activity, and later compute normalized predictivity by dividing model correlation with max ceiling for each channel. Further, the authors  interpret the contribution of both low-level and semantic features in language processing in the brain by looking at individual low-rank components over time and across channels.

**Technical summary:**
This is primarily an empirical study, and its methodology involves the following components:
* *Low-rank tensor decomposition:* The authors use low-rank tensor decomposition in a neural encoding model to predict MEG brain activity. Since MEG data can be processed as  (#Words  x Channels x time lags) and the stimulus representations from LLMs as  (#Words x feature dim), the linear mapping between LLM representations and MEG results weight tensor of shape (time lags x feature dim x channels). With the introduction of low-rank decomposition, the weight tensor decompose into small number of  interpretable components (lag kernel x feature kernel x channel kernel), where the number of components R is the chosen Canonical Polyadic rank of the decomposition.
* *Neural encoding model:* To train neural encoding model, the authors decompose weight tensor and minimise MSE between true and predict MEG brain activity with L2 ridge penalty, and ADAM as optimiser.. For a full-rank model, the authors use ridge regression with channel-specific parameters selected via 6-fold cross-validation.
* *MEG story listening dataset:* The authors use available MEG dataset, where 3 participants listened to 27 naturalistic stories from Moth Radio Hour podcasts over 5 sessions; one story was repeated across sessions and used as held-out test data.

**Experimental design/evaluation:**
* *Noise ceiling:* The authors estimate noise ceiling (CC_max) from 5-repetitions of test story by constructing a sample time series via concatenating windows after word onset of 100 ms. To estimate max ceiling over time, authors subtract CC_max across time windows resulting in getting the excess CCmax given by that time window.
* *Neural encoding performance:* This analysis evaluates whether the low-rank tensor-based neural encoding model shows better encoding performance over full-rank ridge approach. The primary questions are (i) how the encoding performance varies with the increasing rank, and what aspects of language process the low-rank components over time.
* *Controlling low-level features:* This analysis provides whether regressing low-level, control-driven MEG activity from original MEG, and using the residual activity in the low-rank tensor encoding model to see whether high-level semantic emerges over time after controlling low-level features.


**Main findings:**
According to the authors’ interpretation, the main findings are as follows:
* The low-rank tensor encoding model exhibits high-degree of brain predictivity and significantly outperforms the full-rank ridge model across 3 subjects, except at rank-1 for subjects 2 and 3. Encoding performance saturating at small ranks (R=5 to 10).
* Not all low-rank components contributed equally, with some components having large effects while some have rare effects over time.
* A low-rank model with 10 components for all subjects yielded spatial modes over auditory and language areas, while timecourses show peaks around 150–200ms and 300–400ms.
* Examining both positive and negative contexts in driving the activity across the three factor matrices, the authors find that most factors are activated by low-level language features (e.g., sentence starts and ends), while one or two factors respond to semantic features.
* After controlling low-level language features, residual activity components result in most-activating contexts becoming more semantic while 1 to 2 factors respond to low-level.

**Strengths:**

I found this work to have the following strengths:
* *Clarity:* The manuscript is well written and well structured. The pipeline in Figure 1 is easy to follow, and the low-rank tensor decomposition mathematical formulation is well explained by factorizing the weight tensor into three factor matrices: time delay (lag), embedding dimension and channel dimension. Later, the training details of the neural encoding model, and the validation procedure are presented clearly. The results section presents the effectiveness of the low-rank tensor decomposition approach in neural encoding, and interprets the language features that drive each component  as the rank increases.
* *Originality:* The idea of using low-rank tensor decomposition in a neural encoding model for predicting MEG brain activity, while simple, is methodologically novel in MEG encoding. Prior studies in MEG encoding typically use either mTRF (modeling temporal receptive field) approach or flatten channels and delays into a single vector and learn a ridge-regression model (similar to fMRI encoding). With the new encoding approach, it learns fewer parameters while still yielding interpretable components.
* *Significance:* This work is significant in that it contributes to a better understanding of the parallels between language processing in language models and language processing in the human brain over time. It shows that regressing low-level features from MEG activity helps in isolating semantic components, and that low-rank tensor factorization results in learning interpretable components while mapping between model embeddings and brain activity.

**Weaknesses:**

From my perspective, the primary weaknesses of this study arise from the lack of comparison with prior literature, limited evaluation, and limited samples of brain dataset:

* *Lack of comparison with prior literature:* The most significant limitation of current study is the missing direct comparisons and citations of prior works. For instance, Oota et al. 2024 investigated which types of information presented in the text- and speech-based language models truly predict brain activity by regressing low-level textual, speech and visual features from the embeddings and examined the impact of brain alignment before and after removal of low-level features. Further, they showed how both text and speech language models maintain brain-relevant semantics in early sensory and language regions. In contrast, the current study regresses low-level controls from the MEG signal (similar to the residualization approach of Ramakrishna & Deniz on fMRI), but there is no comparison of different low-level features impact, and importantly missing citations.

Oota et al. 2024, Speech language models lack important brain-relevant semantics, ACL-2024

Ramakrishna & Deniz 2021,  Noncomplementarity of information in word-embedding and brain representations in distinguishing between concrete and abstract words. CMCL-2021

* *Small sample size:* Another major limitation is the small number of participants (n=3) in the Moth-Radio-Hour MEG dataset. Although there are other publicly available MEG datasets: MEG-MASC dataset (Gwilliams et al. 2023) (27 participants listening to 4 stories), and the Little Prince dataset in French (d'Ascoli et al. 2024) (58 participants listening, and 46 participants reading), the small sample scale limits the generalizability and statistical power of the findings.

Gwilliams et al 2023. Introducing MEG-MASC a highquality magneto-encephalography dataset for evaluating natural speech processing. Scientific Data

d'Ascoli et al. 2024, Decoding individual words from non-invasive brain recordings across 723 participants

* *Limited model evaluation and lack of low-level controls:*
    * The results and learned components reported only on fixed LLaMA-2-7B layer-3. However, the prior literature on LLM-Brain alignment found that middle to later layers show improved degree of brain alignment [Antonello et al. 2023]. A strong empirical evaluation would include a layer-wise analysis with low-rank tensor factoring to see whether the findings remain held across layers or middle-late layers resulting in high-rank components capturing distinct language processing over time.
    * The procedure for regressing low-level controls and how they perform train-test split is not fully explained in the paper. Furthermore, the low-level controls are regressed from MEG (whereas several studies perform on model features),  which weakens the choice for residual approach on the MEG side.
* *Fixed regularisation parameter and Optimal rank:*
    * For low-rank tensor models, the results are reported using fixed ridge penalty λ=0.1 for all channels because the channels are coupled. This weakens the comparison with the full-rank model. For example, similar to the full-rank model, selecting ridge penalty via 6-fold cross-validation would be stronger and generalizable, as fixed ridge penalty may not be a choice for other datasets.
    * Authors fit a series of low-rank tensor encoding models increasing rank from 1 to 10 and report the performance. The proposed approach should be robust by performing rank-selection via cross-validation approach or validating on more subjects or another MEG dataset.

For a complete and detailed account of both major and minor issues, please refer to the “Questions” section.

**Questions:**

I would like to thank the authors for the interesting low-rank tensor approach in MEG encoding in this work. However, there are several points that I believe require further attention/work. I have divided these into major issues, which should be prioritized, and minor ones, which should be addressed for a strong version of current work.

**Major Comments:**
* *Small sample size:* While I fully understand the complexity associated with empirical research involving human participants, the sample size in this study appears to be very limited. There are additional publicly available naturalistic story-listening MEG datasets with word annotations [Gwilliams et al. 2023]. I strongly encourage the authors to consider performing low-rank tensor decomposition on large sample sizes, and compare the findings with prior approaches. If performing a low-rank approach on all 27 subjects is not feasible, I suggest targeting a subset of the dataset and see whether similar results hold using the same ridge penalty and  low rank (1-10) models. Atleast, the authors provide a clear justification for the current dataset selection and its sufficiency for the conclusions drawn.
* *Comparison with prior works:* Please compare with prior works by using a broader set of low-level textual features (e.g., number of letters, letters, word length+std) and speech features (e.g., MFCC, fbank, powspec, Articulation, number of phonemes, phonemes, and phonological). Both Deniz et al. 2019 and Oota et al. 2024 used the Moth-Radio-Hour podcast (the key difference is fMRI vs. MEG), so these feature sets can be easily transferable. Therefore authors can use all these low-level feature sets and compare the low-level control applied to MEG recordings (compute residuals on brain side) vs.low-level control applied to the stimulus embeddings (residuals from features side). Such a strong empirical validation is crucial for assessing the robustness and reliability of the results.
* *Clarification on Fig 4 and Fig 5:* Both Figs report positive and negative activating contexts. Fig 4 reports without low-level control and interprets the activity in three 1-dimensional factors, whereas Fig. 5 shows after controlling for low-level features. The experimental results demonstrate that “most factors are activated by low level language features, such as end of sentence and start of sentence”. This is a weak finding and the findings are not surprising.
     * Authors should clarify, since this is a listening dataset, how were the sentence boundaries (start/end) created? Did the authors use the same GPT-4 model to create sentence boundaries or boundaries derived through a different preprocessing pipeline?
     * For in-depeth analysis, I recommend the following: Beyond start/end of sentences, punctuations, test part-of-speech categories (nouns, verbs, adjectives), semantic concreteness/abstractness, and other lexical attributes.
     * Examine any semantic-specific relations (e.g., entity-property: Adjective->Noun, Adverb-> Noun) consistently occur in a particular component?
* *Reporting variability across LLM layers:* I recommend authors to perform a layer-wise analysis (at least using middle layer) with a low-rank tensor factoring to test whether the findings remain held across layers or middle-to-late layers resulting in high-rank components capturing distinct language processing over time. If analyzing all layers is not feasible, I suggest selecting the best layer (18) based on Antonello et al. (2023) and performing the analysis there.

Antonello et al. 2023, Scaling laws for language encoding models in fMRI, NeurIPS-2023

* *Clarification on Topomaps in Fig 4:*  The topomaps show high-degree of activity over visual regions despite the task being story listening. Please provide a clear justification on current topomaps.

**Minor Comments/Typos:**
While addressing the following points may not be critical to the paper’s core contributions, doing so would enhance the overall quality.
* Efficiency reporting: I request authors to quantify the efficiency of the low-rank tensor encoding model vs. the full-rank ridge by providing: (i) parameter counts and (ii) compute metrics. A small table would make this clear.
* Line 107 & 161: Canonical Polydiadic -> Canonical Polyadic
* Line 56: I would recommend authors to provide a citation for “Thus, it is difficult to determine which language features in an LLM lead to good predictions of brain activity”. For instance, Oota et al. 2024 investigated this in reading and listening fMRI across text- and speech-based language models.
* Line 90: Authors mentioned that Variance partitioning and Interpreting weights on features are two interpretation methods explored in encoding models	. I recommend also including the residualization approach (Toneva et al., 2022; Oota et al., 2023), as this method is used in several linguistic brain-encoding studies to control features on the model side.
* Please include what each symbol denotes in Fig. 1 within the figure caption for better readability of the pipeline.
* Line 155: N is not defined in that paragraph. Please state that N denotes the feature dimension. Similarly, define n in Eq. 1.
* Line 154: FIR is not abbreviated: Finite Impulse Response (FIR)
* Line 193: 4096-d embeddings -> 4096-D or 4096-dimensional embeddings, same in Fig 4: 1-dimensional or 1-D
* Uniformity is missing: full rank vs. full-rank, low rank vs. low-rank

**General Advice:**
The manuscript presents a methodological novelty in MEG encoding models and a range of experimental design choices for interpreting the features driving the activity that are interesting. However, the current version lacks clear comparison with previous approaches, small sample size and limited experimental evaluation. Adding explicit implications and addressing the above mentioned weaknesses and major comments would make the work stronger.

---

> ### Author Response · Authors · 2025-12-03
>
> We thank the reviewer for their comments. A few clarifications below.
>
>
> We appreciate that other datasets have more subjects, however, our dataset has a lot more material per subject. There is a tradeoff between using a lot of participants and using a lot of stimuli per participants. In the second case, there is enough data to estimate the responses of individuals accurately. Whereas in the datasets with larger numbers of participants, there might not be enough to fit the low rank models. Note that this strategy is also adopted by other large MEG datasets:
>
> Armeni, K., Güçlü, U., van Gerven, M. et al. A 10-hour within-participant magnetoencephalography narrative dataset to test models of language comprehension. Sci Data 9, 278 (2022).
>
>
> The second set of low level controls suggested by the reviewer (pertaining to the audio signal) appear to be more relevant for our speech experiment. We will add the ones we are not already using.
>
>
> Note that for MEG, the layers that have been reported to be more predictive are much lower than the ones reported for fMRI by Antonello et al. Please see:
>
> Zhou, Y., Liu, E., Neubig, G., Tarr, M., & Wehbe, L. (2024). Divergences between language models and human brains. Advances in neural information processing systems, 37

---

### Official Review · Reviewer_SBZ9 · 2025-10-30

**Soundness:** 2
**Presentation:** 2
**Contribution:** 2
**Rating:** 2
**Confidence:** 3

**Summary:**

The authors propose a low-rank tensor regression model to analyze Magnetoencephalography (MEG) data from subjects listening to naturalistic narrative stories. The core of the method is to model the high-dimensional Finite Impulse Response (FIR) filter—which maps LLM embeddings to MEG channel activity over time—as a low-rank Canonical Polydiadic (CP) tensor. This decomposition factors the model's weights into a small number of components, each comprising separate 1D vectors for time delays, embedding dimensions, and MEG channels.

**Strengths:**

The idea of building a low-rank tensor encoding model that captures low-level features is interesting. The experiments also confirm its effectiveness compared to traditional ridge regression.

**Weaknesses:**

1. The experiments are not satisfying. The authors fall short in investigating low-rank tensor encoding model in the context of other cognitive signals (e.g. fmri). Besides, the study's experiment on MEG data only involves 3 patients. Given the known variability across individual brain response, N=3 is an insufficient sample size to make broad claims. Previous studies used fmri dataset with hundreds of subjects.

2. There lack experiment and analysis of computational cost between traditional ridge regression and the proposed low-rank tensor encoding model (e.g. the time cost comparison).

3. The analysis primarily relies on cases to prove low-level controls improve semantic features. A proper evaluation metric is needed.

4. The performance comparison in Section 3.2 is potentially flawed. The full-rank ridge regression baseline used 6-fold cross-validation to select channel-specific regularization parameters ($\lambda_c$). In contrast, the proposed low-rank model used a fixed ridge penalty of 0.1 for all channels, justified as "near the full regression average." This is not a fair comparison.

**Questions:**

1. In line 152-157 the authors mention "Unlike fMRI, whose time course is slower than the usual pace of spoken words, MEG is fast enough to resolve brain activity during the processing of each word.". I do not understand why the authors mention this, because FIR model is also used to preprocess fmri data. The expression is confusing.

---

> ### Author Response · Authors · 2025-12-03
>
> We thank the reviewer for their comments. A few clarifications below.
>
>
> We do not understand the point of the reviewer in comparing our methods to fMRI. fMRI has a very different spatiotemporal profile, and it is not possible to model the signal at the tens of milliseconds level, and the subword level, the way it is possible in MEG. While a tensor-factorized model could still be used, it would typically be less interesting, because the temporal dimension would correspond to FIR lags as suggested by the reviewer. These FIR lags would reveal the temporal lags of the FIR for different factors (we don’t have a reason to assume they would be different), which is more of a brain biophysical property. In contrast, the temporal profiles in MEG instead correspond to delays in brain responses, not in hemodynamic responses. Thus, in MEG we can use this method to look at components that are processed early on (such as 100ms after word onset) or later one (such as 300ms after word onset), and thus this contributes to our understanding of how these dynamical processes occur in the brain.
>
>
> As stated above, allowing the model to choose a different penalty for each sensor tends to improve the average accuracy. Thus, the full-rank baseline is at an advantage compared to our low-rank model, and the low-rank model still outperforms it.

---

### Official Review · Reviewer_67ce · 2025-11-01

**Soundness:** 3
**Presentation:** 3
**Contribution:** 3
**Rating:** 6
**Confidence:** 4

**Summary:**

This paper proposes a low-rank tensor regression framework for MEG-based speech encoding models. The method factorizes the encoding weight tensor into interpretable components corresponding to time, language embeddings, and brain channels, enabling both improved predictive performance and interpretability compared to standard ridge regression. Applied to a naturalistic MEG dataset (participants listening to Moth Radio Hour stories), the approach demonstrates that low-rank models not only yield higher encoding accuracy but also reveal distinct temporal and spatial neural components reflecting different stages of speech comprehension (e.g., early auditory vs. later semantic processing). After regressing out low-level acoustic and structural language features, the remaining components exhibit stronger sensitivity to semantic information, suggesting that the method captures hierarchically organized linguistic processing in the brain.

**Strengths:**

- The **low-rank tensor formulation** is well-motivated, elegant, and appropriately implemented.
- Empirical results convincingly show that the low-rank inductive bias improves both **predictive performance** and **interpretability**.
- The **low-level control analysis** is a novel point, as it helps disentangle acoustic from semantic representations — an important concern in recent brain–LLM alignment studies.

**Weaknesses:**

- **Quantification of interpretability:** The interpretive claims (e.g., that components correspond to distinct linguistic or cognitive processes) rely mostly on qualitative examples and visualizations. Quantitative or systematic metrics for interpretability (e.g., linguistic feature correlations) would strengthen the argument.
- **LLM comparison :** I think that the analysis of the components and their positive/activating contexts in Figure 4 should also be done on LLM embeddings too. This would ensure that the linguistic feature distinction that we see from low-rank models is not simply due to the fact that the inputs (LLM embeddings) already contain such information.
- **Model training inconsistency:** It is unclear how the full-rank model was trained, as opposed to the low-rank model, which the author specifies as being trained using SGD. Could the author indicate the training method for the full-rank model? If they differ, then this could confound performance comparisons and should be justified or controlled for.
- **Reference mismatch:** The citation to *Vattikonda et al. (2025)* as an example of low-rank encoding is misleading — that paper applied LoRA on speech model as a fine-tuning mechanism, not a low-rank encoding structure per se.
- The current study is **single-subject focused** in analysis; cross-subject generalization or shared subspace modeling would add more weight to the neuroscientific claims.

**Questions:**

1. When performing PCA on LLM embeddings, was this transformation fit on the entire moth radio corpus (including test data)? If so, this could introduce **train-test leakage**.
2. Could the authors clarify how **excess CCmax** is defined mathematically? Is it simply the difference between windowed and global CCmax values? If so, please clarify the use of “excess” as the name. Also, since the time window CCmax is computed using a smaller number of timepoints (less samples), I believe it is important to also calculate their confidence intervals.
3. Have the authors considered quantifying the linguistic selectivity of each factor (e.g., correlation with word frequency, surprisal, or syntactic depth)?

---

> ### Author Response · Authors · 2025-12-03
>
> We thank the reviewer for their suggestions. Some responses below:
>
>
> The PCA was only performed on the training data. The excess CCmax is indeed the difference between windowed and global CCmax values. We thank the reviewer for the suggestion of quantifying the interpretation more using linguistic features, and of interpreting the LLM embeddings to verify their content. We will integrate these in the manuscript.

---

### Official Review · Reviewer_G5Sg · 2025-11-04

**Soundness:** 3
**Presentation:** 3
**Contribution:** 3
**Rating:** 6
**Confidence:** 3

**Summary:**

This paper presents a low-rank tensor regression method for building encoding models of MEG brain activity during language comprehension. Core idea is to model the linear filter, which maps LLM embeddings to neural signals, as a 3D tensor with dimensions for time, embedding features, and MEG channels.The authors constrain this tensor using a low-rank canonical polyadic decomposition. This approach models the filter as a sum of rank-1 components, where each component is the outer product of a temporal profile, a spatial map across channels, and a weight vector over the embedding features.

The authors apply this method to an MEG dataset of subjects listening to narrative stories and report two main contributions. First, they show that the low-rank model achieves higher prediction accuracy than a standard full-rank ridge regression model, suggesting that the low-rank structure is a useful inductive bias. Second, they demonstrate the model's interpretability by analyzing the individual components. They find that the model separates distinct language processing features, with many components capturing low-level features like sentence onsets or punctuation, while others capture more semantic content. To isolate these semantic components, they fit the model to the residual signal after regressing out low-level audio and word-onset features, showing that the resulting components are more clearly semantic.

**Strengths:**

The paper's primary contribution is the application of low-rank tensor decomposition to model the MEG encoding filter. This decomposition of the filter into separate, interpretable factors for time, channel, and embedding features is elegant and provides a path to understanding the model's components.

The authors demonstrate that this model outperforms the standard full-rank ridge regression baseline in prediction accuracy, and that it offers a way to disentangle signals present in MEG. The control analysis, where low level audio and linguistic features are regressed out, is interesting. The analysis seems to support the claim that the method can isolate higher order semantic processing, a key goal in the field.

**Weaknesses:**

The choice of a canonical polyadic tensor decomposition is a very strong inductive bias. This model forces each component to be a strict outer product of one time factor, one channel factor, and one embedding factor, assuming a simple one to one mapping that may not be neurally plausible. A justification for this choice over a more flexible tucker decomposition, which can model interactions between factors via a core tensor, is missing.  This issue is amplified by an inconsistent comparison to the baseline. The full rank model benefits from a careful per channel search for its regularization penalty, while the proposed low rank model uses a single fixed penalty. This methodological difference confounds the central claim that the low rank structure itself is the source of the performance gain.

The approach to isolating semantic features is also a bit problematic. The authors use a two stage process, first fitting a model to low level controls and then fitting their tensor model to the residuals. This stacked regression method is sensitive to the order of operations and can misattribute any shared variance between the control features and the LLM embeddings. Finally, the interpretation of the resulting semantic components is based on an anecdotal inspection of the most activating text snippets. This might be insufficient for characterizing a high dimensional embedding factor.

**Questions:**

1. Can you elaborate on the choice of the canonical polyadic decomposition? This model imposes a strict multiplicative structure on the time, channel, and embedding factors. Did you explore a more flexible tucker decomposition, which might better capture interactions via a core tensor, and how did it compare in performance and interpretability?

2. Could you clarify the rationale for using a fixed ridge penalty for the low-rank model while the full-rank baseline used a per-channel cross-validated penalty? How can we be sure the reported performance gain is due to the low-rank inductive bias itself and not this difference in regularization strategy?

3. The paper uses a two-stage process to control for low-level features by fitting the model to the residuals. This stacked regression can misattribute variance that is shared between the controls and the LLM embeddings. Have you compared this to a joint model that fits all features simultaneously, and do the semantic components that emerge differ?

4. The interpretation of the embedding factors relies on inspecting a few top-activating text snippets. Have you considered quantitatively correlating the learned embedding factor vectors with established linguistic probes to confirm they are capturing specific semantic or syntactic properties?

---

> ### Author Response · Authors · 2025-12-03
>
> We thank the reviewer for their comments. A few clarifications below.
>
>
> The rigid choice of the canonical polyadic decomposition is used as a prior on a specific factor. The idea is that only specific regions would be involved in representing that factor and that this representation would occur with a specific temporal signature.
>
>
> Allowing the model to choose a different penalty for each sensor tends to improve the average accuracy. Thus, the full-rank baseline is at an advantage compared to our low-rank model, and the low-rank model still outperforms it.
>
>  After regressing out our controls (we do not use a ridge penalty to do so), the resulting residuals are orthogonal to the controls. Thus, this should take into account any interaction with the shared variance with LLM embeddings, since the data is no longer predictable from the low level controls. As seen in figure 4, not regressing out these controls results in many components focusing on low level features.

---

### Meta-Review · Area_Chair_PG9F · 2026-01-06

**Summary:**

The reviewer’s concerns were centered around limitations in the empirical evaluation.

1. The brain dataset used includes only three subjects.

2. The study uses a single LLM model (LLaMA-2-7B, layer 3) and does not report results for other models or layers.

3. The analysis is single-subject–focused, and generalization across subjects is not evaluated.

4. The study lacks comparisons with prior work.

**Reviewer Concerns:**

The authors’ response was minimal and only partially addressed the concerns outlined above.

1. The authors argue that the dataset contains “a lot more material per subject,” which does not directly address the stated concerns.

2. The authors cite a paper by Zhou et al. supporting the use of an early LLM layer, but do not otherwise address the concern by providing additional empirical evidence.

3. Not addressed.

4. Not addressed.

**Reviewer Scores:**

After carefully reading the reviews and the author response, I highly doubt that any reviewer will change their score.

---

### Decision · Program_Chairs · 2026-01-26

Reject